# Inter-Individual Variation in DNA Methylation Patterns across Two Tissues and Leukocytes in Mature Brahman Cattle

**DOI:** 10.3390/biology12020252

**Published:** 2023-02-05

**Authors:** Emilie C. Baker, Audrey E. San, Kubra Z. Cilkiz, Brittni P. Littlejohn, Rodolfo C. Cardoso, Noushin Ghaffari, Charles R. Long, Penny K. Riggs, Ronald D. Randel, Thomas H. Welsh, David G. Riley

**Affiliations:** 1Department of Animal Science, Texas A&M University, College Station, TX 77845, USA; 2Texas A&M AgriLife Research, College Station, TX 77845, USA; 3Texas A&M AgriLife Research & Extension Center at Overton, Overton, TX 75684, USA; 4Department of Computer Science, Prairie View A&M University, Prairie View, TX 77446, USA

**Keywords:** Brahman, methylation, prenatal stress, variability

## Abstract

**Simple Summary:**

Epigenetic modifications such as DNA methylation can influence gene expression and phenotype. Variation in DNA methylation patterns between individuals may contribute to phenotypic variation. The object of this study was to quantify the inter-individual variation in DNA methylation patterns of the anterior pituitary, amygdala and leukocytes harvested from two groups of Brahman females, one prenatally stressed and one control. There was little overlap between the sites and areas that exhibited high inter-individual variation between the two groups. The interaction between the prenatal environment and cow genotype could be responsible for the differences in location of the variation. The variation also appeared to be tissue specific, providing support for DNA methylation’s role in tissue specific gene expression. Genes that displayed high variation in methylation are active in biological pathways important to immune response, hormone production and behavior. This was the first characterization of the inter-individual variation of DNA methylation in somatic cells of beef cattle. Further research characterizing how methylated regions interact with gene expression and the environment may give useful insight into how cow performance is affected.

**Abstract:**

Quantifying the natural inter-individual variation in DNA methylation patterns is important for identifying its contribution to phenotypic variation, but also for understanding how the environment affects variability, and for incorporation into statistical analyses. The inter-individual variation in DNA methylation patterns in female cattle and the effect that a prenatal stressor has on such variability have yet to be quantified. Thus, the objective of this study was to utilize methylation data from mature Brahman females to quantify the inter-individual variation in DNA methylation. Pregnant Brahman cows were transported for 2 h durations at days 60 ± 5; 80 ± 5; 100 ± 5; 120 ± 5; and 140 ± 5 of gestation. A non-transport group was maintained as a control. Leukocytes, amygdala, and anterior pituitary glands were harvested from eight cows born from the non-transport group (Control) and six from the transport group (PNS) at 5 years of age. The DNA harvested from the anterior pituitary contained the greatest variability in DNA methylation of cytosine-phosphate-guanine (mCpG) sites from both the PNS and Control groups, and the amygdala had the least. Numerous variable mCpG sites were associated with retrotransposable elements and highly repetitive regions of the genome. Some of the genomic features that had high variation in DNA methylation are involved in immune responses, signaling, responses to stimuli, and metabolic processes. The small overlap of highly variable CpG sites and features between tissues and leukocytes supports the role of variable DNA methylation in regulating tissue-specific gene expression. Many of the CpG sites that exhibited high variability in DNA methylation were common between the PNS and Control groups within a tissue, but there was little overlap in genomic features with high variability. The interaction between the prenatal environment and the genome could be responsible for the differences in location of the variable DNA methylation.

## 1. Introduction

Epigenetic mechanisms influence gene expression without changing the underlying DNA sequence. One epigenetic modification is DNA methylation, which typically occurs through the addition of a methyl group to the 5′ carbon of the nitrogenous base cytosine in mammals [1]. Most DNA methylation occurs at cytosine-phosphate-guanine (CpG) sites and clusters of CpG sites known as CpG islands [2]. Methylation can influence gene expression by changing the accessibility of the gene to the needed transcription factors and influencing the splicing of transcripts. Considering the influence epigenetic modifications have on gene expression, variation among DNA methylation patterns within individuals can contribute to phenotypic variation [3]. Inter-individual variation of DNA methylation has been observed in different human populations as early as the germ cell stage [4,5]. Comparison of methylation patterns of neutrophils from a group of healthy individuals identified over 12,000 inter-individual variable fragments throughout the autosomes [6]. Similar patterns were observed in peripheral blood monocytes in humans [7]. Variation in DNA methylation patterns throughout the genome could contribute to variations in behavior, immune response, growth, and responses to the environment and drug treatments. 

There is a strong genetic component to inter-individual DNA methylation variation [8,9]. Single nucleotide polymorphisms at CpG sites can directly lead to variation in methylation patterns and affect expression levels by altering recognition sites for transcription factors and DNA methyltransferases [10,11]. The amount of inter-individual variation is often tissue dependent. Human neuron cells had higher inter-individual variation in DNA methylation patterns relative to non-neuron cells [12]. Time and environment also influence the inter-individual variation of DNA methylation, as inter-individual variation tends to increase over time [13,14]. In twins, it was found that a large contributor to inter-individual variation in DNA methylation patterns was environmental factors [15]. The interaction between genetics and different uterine environments (maternal smoking, maternal depression, maternal body mass index) was the best explanation for 75% of the variably methylated regions found in neonates [16]. 

Prenatal and early life stress induced alterations in DNA methylation patterns of the offspring in cattle, and some of those alterations persisted later in life [17,18,19]. The prenatal environment and stressors can explain a portion of the inter-individual variability of methylation patterns in umbilical cord tissue and whole blood [15]. However, less is known about the effect of the prenatal environment on the inter-individual variation within tissues involved in stress response, or on the magnitude of inter-individual variation. In humans, the amount of inter-individual variation of methylation levels in the gene nuclear receptor subfamily 3 group C member 1 did not differ between those who had experienced traumatic events and those who had not [20]. 

Quantifying the variation in natural populations has aided in determining the relevance of variation in DNA methylation patterns and the effect it has on phenotypic variation. Understanding the variation between healthy individuals can aid in understanding how a treatment or stressor affects normal methylation patterns [5]. The variation between individuals is also important for statistical analysis. Variability can also make it difficult to identify significant associations within expression data. The variation must be considered when selecting the proper sample size and analysis methodology [21]. 

There has been a single study on the inter-individual variation of DNA methylation in cattle focusing on methylation patterns in spermatozoa harvested from Holstein bulls [22]. The methylome has the potential to influence important production aspects such as susceptibility to disease, adaptability to stressors such as heat stress, fertility and even food intake. Variability in DNA methylation patterns between cows could result in performance differences that impact profit. The inter-individual variation in DNA methylation patterns has yet to be investigated in mature female cattle. Little is known about how inter-individual variation in DNA methylation differs from tissue to tissue or depending on how a stressor affects the variation in mature female cows. Brahman cattle are better adapted to subtropical and tropical climates than other cattle breeds. The heat and insect tolerance exhibited by the Brahman breed makes them an essential part of beef cattle production in the southern United States and areas with warmer climates. Thus, this project aimed to classify the inter-individual DNA methylation variation in tissues and leukocytes of mature Brahman cows through (1) visualization of the range of methylation at sites across the genome, (2) identification of genomic features with high variability in methylation, and (3) comparison of those results across tissues and leukocytes. 

## 2. Methods & Materials 

All procedures were done in compliance with the Guide for the Care and Use of Agricultural Animals in Research and Teaching [23] and its earlier versions, and approved by the Texas A&M AgriLife Research Animal Care and Use Committee. 

### 2.1. Animal Procedures

An in-depth description of the experimental design is in Littlejohn et al. [17]. In brief, a group of pregnant Brahman cows was transported for 2 h durations at days 60 ± 5, 80 ± 5, 100 ± 5, 120 ± 5, and 140 ± 5 of gestation. These dams exhibited increased vaginal temperature, shrink, and increased serum cortisol and glucose in response to the transportation events, confirming a physiological stress response [24]. A non-transport group was maintained as a control. Both groups were managed under the same environmental and nutritional conditions at the Texas A&M AgriLife Research & Extension Center at Overton, TX (32.27° N, −94.98° W). Twenty-one heifer calves were born from the transported cows (PNS), and 18 heifer calves were born to cows that had not been transported (Control). The heifer calves were exposed to bulls for mating at 1 year of age and annually thereafter. From those females that remained at 5 years of age, 6 PNS and 8 Control cows were slaughtered, and the amygdala and the anterior pituitary glands were collected. At the time of harvest, 10 mL peripheral blood were collected via a vacuum tube venipuncture for isolation of leukocytes. The amygdala and anterior pituitary gland were chosen because of their importance in hormone production and stress response. Leukocytes were chosen for the analysis as a sample set originating from outside of the brain.

### 2.2. Sample Preparation & DNA Extraction 

The anterior pituitary and amygdala tissues were cut and weighed to 20 mg. All tissue samples were snap-frozen with liquid nitrogen and then stored at −80 °C until analysis. Before DNA isolation, tissue samples were digested with proteinase K in a water bath at 56 °C, and the GeneJET Genomic DNA Purification Kit (Thermo Scientific, Waltham, MA, USA) DNA purification protocol was used to isolate DNA from the anterior pituitary and amygdala samples. The purified DNA samples were quantified with a NanoDrop Spectrophotometer (NanoDrop Technologies, Rockland, DE, USA) and stored at −80 °C until further analysis.

Blood samples were centrifuged at 2671× *g* for 30 min at 6 °C. The white blood cell layer was then isolated and placed into 2-mL nuclease-free microcentrifuge tubes. The white blood cell layer was washed repeatedly with red blood cell lysis buffer solution until a clean cell pellet was produced. A phenol-chloroform extraction procedure was used to extract DNA from the isolated white blood cell pellet as described by Littlejohn et al. (2018). In brief, the white blood cell pellets were placed in an extraction buffer (100 mM NaCl, 10 mM Tris, 1 mM EDTA, pH 7.5) and 10 mg/mL proteinase K and 20% SDS was added for proteinase K digestion. Samples were then incubated and extracted twice with an equal volume of phenol:chloroform:isoamyl alcohol (25:24:1), and twice with an equal volume of 1-bromo-3-chloropropane (substituted for chloroform). The DNA was precipitated by the addition of 10% 3 M sodium acetate (pH 5.2) and one volume of isopropanol to the solution. The isolated purified DNA was suspended in 150–200 μL TE buffer (10 mM Tris, one mM EDTA, pH 8.0) and stored at –80 °C. 

### 2.3. DNA Methylation Analysis 

Purified DNA from each tissue was submitted to Zymo Research (Irvine, CA, USA) for reduced representation bisulfite sequencing analysis. First, the DNA was digested with 60 units of TaqαI followed by 30 units of MspI and then purified with DNA Clean & Concentrator^TM^. Adapters containing 5′-methyl-cytosine were then ligated to the fragments. Adapter-ligated fragments of 150 to 250 bp and 250 to 350 bp were recovered using the Zymoclean^TM^ Gel DNA Recovery KitRe and then ligated to the purified DNA fragments. Recovered fragments were then bisulfite-treated using the EZ DNA Methylation-LightningTM Kit. An Illumina HiSeq base calling was used to identify and sequence reads from the bisulfite-treated libraries. After the raw FASTQ files were quality trimmed and assessed (TrimGalore 0.6.4, FastQC 0.11.8), they were aligned to the *Bos taurus* genome (ARS-UCD1.2; Rosen et al., 2020) using Bismark 0.19.0 (Babrahman Bioinformatics, Cambridge, United Kingdom). Alignment produced binary alignment map (BAM) files. Methylated and unmethylated read totals for each site were called using MethylDackel 0.5.0 (Zymo Research).

### 2.4. Statistical Analysis 

Two approaches were used to analyze the variation in DNA methylation patterns within each prenatal treatment group and each tissue. 

#### 2.4.1. Genome-Wide Inter-Individual Methylation Variation

Methylated and unmethylated read counts for all samples in each tissue were imported into the edgeR packages (Version 3.40.0) from Bioconductor [25]. Sites were then filtered using the criterion of having at least 5× coverage across all samples within a group. Beta (β) values, which estimate methylation levels using the ratio of reads mapped between methylated and unmethylated alleles plus a normalizing factor of 1000, were calculated in each sample at the sites that passed filtering [26]. Pearson correlation coefficients between animals within a group were calculated using the β values of the CpG sites across the genome. To visualize the variability of methylation at each site within each group, the β values were used to calculate a site’s inter-individual β value range (IBR) by subtracting the smallest β value from the largest β value at each site in the groups. Inter-individual β value ranges have been used before to visualize the inter-individual variation in DNA methylation patterns in blood mononuclear cells and buccal epithelial cells [27].

The variance and standard deviation of the β values of sites were used to identify CpG sites that had high variation within a group in a tissue. Within each tissue and group, the standard deviations at the CpG sites were filtered to find the sites that had SD ≥0.1. The variance of the β values at each site was calculated, and the mean variance was calculated for each group in each tissue. Sites with variance greater than the tissue mean variance were identified with chi-square (χ^2^) tests (*FDR* < 0.001), after correction for multiple comparisons, per Benjamini and Hochberg [28]. Sites that passed the filters were input into the UCSC Data Integrator tool [29] to identify the genomic regions of the sites. 

#### 2.4.2. Genomic Feature Inter-Individual Methylation Variation

The BAM files provided from the Zymo analysis were read into the analysis program SeqMonk (Babrahman Bioinformatics, Cambridge, United Kingdom). Genomic features were defined for variation analysis. Four different genomic features were defined: gene bodies, CpG islands, CpG shores (2000 bp upstream and 2000 bp downstream of CpG islands), and promoter regions (1000 bp upstream of transcription start site and 500 bp downstream of the transcript start site). After each feature type was defined, a bisulfite feature methylation pipeline (SeqMonk) was applied with the requirement that the sites within the feature have at least 5× coverage. Estimates of percentage methylation for each cytosine within features were averaged to give an overall methylation value. Features with no methylation values were marked as null and filtered out of the statistical analyses.

The standard deviation of each feature within a group was calculated using the overall methylation value. The features were then filtered using the variance intensity difference statistical test (SeqMonk) which is used to identify high or low variance values within a replicate set. From the standard deviations of all features, a subset was selected at random to construct a distribution for comparison. For these analyses, the number of standard deviations selected was equal to 1% of the total number of features. Each feature’s standard deviation was tested to identify the probability of its value occurring outside of the constructed distribution. False discovery rate methodology [26] was then applied to adjust for multiple comparisons. The features with *FDR* ≤ 0.05 were considered to have a high standard deviation relative to other features, and therefore highly variable in their group. Biological pathways and functions corresponding to the highly variable features were identified using PANTHER software [30].

## 3. Results

### 3.1. Genome-Wide Inter-Individual Methylation Variation 

#### 3.1.1. Pearson Correlation and Inter-Individual β Value Range

The numbers of CpG sites that passed filtering were different for each tissue: 63,255 in the amygdala, 1,662,183 in the anterior pituitary gland and 526,816 in the leukocytes. Pearson correlation coefficient estimates for the β values at those sites across the genome were high (*r* ≥ 0.80) between samples in each tissue (Figure 1). The strongest correlations were between amygdala samples, with similar values among samples in the Control group and in the PNS group (Figure 1A,B). The Pearson correlation coefficients between samples within the PNS and Control for the anterior pituitary gland (Figure 1C,D) were lower than the amygdala, but similar to the correlation values between leukocyte samples (Figure 1E,F). The mean IBR was 0.0356 for the amygdala samples from the Control group (Figure 2A). The PNS group had a similar mean IBR of 0.0354 (Figure 2B). The sites within the pituitary gland had the largest mean IBR (Figure 2C,D), closely followed by the mean IBRs from the leukocytes (Figure 2E,F). While each had a distinct distribution of IBR at CpG sites across the genome, both tissues and leukocytes in each group had bimodal distributions. 

#### 3.1.2. Standard Deviation of β Values

Like the pattern observed in the inter-individual β value range (IBR) values, the tissues and leukocytes showed distinct differences in distribution of the SD of β values of CpG sites across the genome (Figure 3). In the amygdala, the PNS group had slightly more CpG sites than the Control group, with SD ≥ 0.1. Seventy-four sites that passed filtering with SD ≥ 0.1 in the PNS group were found in the Control. For the leukocytes, the PNS group had more sites with SD ≥ 0.1 relative to the Control group (Table 1). Of the sites identified to have SD ≥ 0.1, in the leukocytes 152 were common to both groups. The anterior pituitary gland had the most CpG sites, with SD ≥ 0.1 (Table 1). Of those sites, 550 had a standard deviation of 0.1 or greater in both the PNS and Control groups. Pairwise comparison between tissues and leukocytes revealed similar numbers between each pair, with the anterior pituitary and amygdala sharing the most sites (Figure 4A). The tissues and leukocytes from the PNS group shared slightly more sites across tissues. Pairwise, the tissues and leukocytes shared slightly more than observed in the Control group, and again the anterior pituitary and amygdala shared the most sites (Figure 4B). 

The sites with a SD ≥ 0.1 in both tissues and leukocytes in both groups were located in various types of repetitive elements of the genome, including short and long interspersed nuclear elements (SINE, LINE), ribosomal RNA sequences (rRNA), and satellite elements (Figure 5). There was no consistent pattern regarding what type of repetitive element had the most sites located within it across the tissue types. The repetitive element type with the most sites located within it was more consistent within tissue. In the anterior pituitary gland, LINEs and SINEs had the most CpG sites, with SD ≥ 0.1 in both the PNS and Control groups, while LINEs and long terminal repeats (LTRs) had the most sites located within them in the PNS and Control amygdala samples. In the Control and PNS leukocytes, LTRs had the most sites located within them; however, while the Control leukocytes had few sites (*n* = 4) located within rRNA sequences, the PNS leukocytes exhibited 86 sites with SD ≥ 0.1 located within rRNA sequences. 

#### 3.1.3. Chi-Square Test for the Variance

The anterior pituitary in both the PNS and the Control groups had the most sites, with a β value variance that was different from the mean variance, and the amygdala had the least (Table 1). The leukocytes from the PNS group had more significant sites (22,574) than the Control group. In the amygdala and the anterior pituitary, the Control group had more significant sites: 977 and 28,863, respectively. The tissues and leukocytes from the Control group shared 2415 variable (*FDR* ≤ 0.001) sites. The Control leukocytes and anterior pituitary gland shared the most sties, closely followed by the anterior pituitary and amygdala (Figure 4C,D). In the tissues and leukocytes from the PNS group, 2919 variable (*FDR* ≤ 0.001) sites were shared across all three. The anterior pituitary and amygdala shared the most sites, 10,837, while the other tissue and leukocyte pairs shared considerably fewer. The PNS and Control group for each tissue shared more sites with *FDR* ≤ 0.001 relative to the sites with SD ≥ 0.1 (Table 1). For the amygdala, the PNS and Control groups shared 83.63% of the significant sites in the Control and 90.74% of the significant sites in the PNS group. The majority of significant sites were shared between the PNS and Control group in the anterior pituitary. The leukocytes from the PNS and Control group shared 26,963 variable sites, which was only 44.10% of the significant sites in the PNS and 69.90% of the significant sites in the Control.

### 3.2. Genomic Feature Inter-Individual Methylation Variation

The number of features analyzed before filtering for null values varied for each feature type: 26,863 genes and promoter regions, 22,188 CpG islands, and 44,376 CpG shores. 

As in the genome-wide variation analysis, the amygdala had the smallest number of features tested after removing features with null values for methylation levels (Table 2 and Table 3). The amygdala also had the fewest features identified to have high variation in both the PNS group and the Control group. The anterior pituitary had the most genomic features that had significant variation in DNA methylation patterns, followed by the leukocytes. The significant features for both tissues and the leukocytes made up a small fraction of the total features tested, with the amygdala features ranging from 0.398% to 0.897%. While the anterior pituitary had the most features that were highly variable for the PNS and Control, they were a small portion of the total features that were tested (Table 2 and Table 3). 

The magnitude of variable methylated features identified in the PNS and Control groups was relatively similar, but few features were highly variable in more than one tissue or leukocytes (Table 4). In the Control group, only one gene had high variability in both tissues and leukocytes, guanylate cyclase activator 2B (*GUCA2B*, *Bos taurus* chromosome (BTA) 3:104,021,061–104,024,316). No promoter had variable methylation in the two tissues and the leukocytes; the anterior pituitary shared 15 with the leukocytes and four with the amygdala (Appendix A). The leukocytes only shared three variable methylated promoter regions with the amygdala. A similar trend was observed in the CpG islands, with no features shared across all three, and few shared between two tissues and leukocytes (Appendix A). No CpG shores were common to all three nor pairs of tissues and leukocytes (Appendix A). 

Again, only two genes exhibited high methylation variation in all three PNS group tissues (ENSBTAG00000036102, BTA 7:68,625,875–68,685,772; ENSBTAG00000051147, BTA 4:106,051,439–106,054,690). Both genes are labeled as novel genes in the Ensembl ARS-UCD1.2 reference genome [31]. Pairwise comparisons of tissues and leukocytes indicated minimal overlap in genes with variable methylation (Figure 6). A single promoter region showed variable methylation across tissues and leukocytes: the promoter region for Synaptotagmin 4 (*SYN4*) (Appendix A). No CpG islands were variably methylated across all tissues and leukocytes in the PNS group; however, 25 CpG islands were variably methylated in both the anterior pituitary gland and leukocytes (Appendix A). As in the Control cows, no variable CpG shore methylation sites were shared between the three or between the pairs of tissues and leukocytes (Appendix A). 

The melanin-concentrating hormone receptor 2 (*MCHR2*, BTA 9:49,923,127–49,948,405) gene had high variability in DNA methylation within the amygdala. The gonadotropin-releasing hormone receptor (*GNRHR,* BTA 6:83,434,759–83,452,201) and the prolactin-related protein 3 (*PRP3*, BTA 23:35,089,013–35,100,774) had highly variable DNA methylation patterns in the anterior pituitary. Interleukin 36 alpha (*IL36A*, BTA 11:46,700,982–46,704,288) and C-X-C motif chemokine receptor 2 (*CXCR2,* BTA 2:106,185,020–106,192,570) were highly variable in the PNS group. Pathway analysis revealed that many genes and promoter regions with variable methylation in the pituitary gland and the amygdala, such as the ones listed above, are involved in biological pathways such as signaling, responses to stimuli, and metabolic processes (Appendix A). The products of *PRP3* and *GNRHR* are active in responses to stimuli and are integral parts of the gonadotropin-releasing hormone receptor pathway. Genes and promoter regions of genes that were highly variable in the leukocytes were involved in immune response (Table 5). Genes such as *IL36A* and *CXCR2* are involved in the immune response process, the inflammation mediated by chemokine and cytokine signaling pathway (Appendix A). 

## 4. Discussion

Methylation patterns in prenatally stressed Brahman cattle was first presented by Littlejohn et al. [17]), Baker et al. [18], and Cilkiz et al. [19]. In the leukocytes harvested at 28 days of age from the same animals described in this project, there were vast differences in DNA methylation patterns between the prenatally stressed group and the control. Many of the differentially methylated cytosines were located within regulatory genes active in hormone production, immune response, and development [17,18]. Differences in methylation of the leukocytes between the two groups diminished between 28 days and 5 years of age [19]. Methylation patterns varied within each group at both time periods. The individual variability in these patterns has been reported in humans [6] and in bovine sperm [22]. However, methylation patterns differ greatly between species, sexes, and tissues, suggesting the variation observed in either would not adequately portray the inter-individual variation in bovine soma cells. This study utilized methylation data in order to quantify the inter-individual variation in DNA methylation in peripheral blood leukocytes, brain (amygdala), and endocrine (anterior pituitary gland) tissue samples. DNA from the anterior pituitary contained the most variably methylated features, followed by DNA from leukocytes. Both had substantially more methylation variability than DNA from amygdala tissue. The same pattern was observed in the number of mCpG sites with β value SD ≥ 0.1, as well as sites with *p* ≤ 0.001 for the χ^2^ test for the variance. While correlation between samples in the tissues and leukocytes was large and positive, the samples from the amygdala showed the strongest correlations, in both the PNS and Control groups, out of the three. The amygdala had the lowest number of sites and features identified to have higher variance estimates. Cell type and heterogeneity within a sample can influence the variability of methylation [26,32]. Leukocytes consist of numerous cell types, such as monocytes, leukocytes, and neutrophils, which can each have different methylation profiles [33]. The anterior pituitary gland and the amygdala also contain numerous cell types [34,35]. The varying cell types in a sample of one tissue can contribute to the high variation relative to another. Isolation of a singular cell type for inter-individual variation analysis could prevent the confounding effect of different methylation profiles in each cell type [6].

In the amygdala, pituitary tissues and leukocytes, CpG islands and CpG shores were identified with high variation in methylation patterns. Sites with SD ≥ 0.1 and *p* ≤ 0.001 were also located within CpG islands throughout the genome. Increased inter-individual variation within CpG shores was identified in human blood and cerebellum samples [36,37]. High levels of inter-individual variation within CpG islands have also been reported in human germ cells [4]. The variability of methylation patterns observed here in the two tissues and leukocytes does not follow the general trend of CpG islands being largely and consistently unmethylated [38]. However, the number of islands identified to have variable methylation makes up a very small percentage of the CpG islands tested. 

Variation in DNA methylation of genes and promoter regions can result in variability in gene expression [39]. Variable methylation in the dopamine receptor D4 contributes to variations in gene expression and natural variation in bird behavior and personality [40]. Variable methylation patterns within the gene pro-opiomelanocortin in humans may affect body weight regulation [41]. The receptor coded by *MCHR2* is influenced by a melanin-concentrating hormone in the amygdala, and can control feeling and motivational behavior [42]. The gonadotropin-releasing hormone receptor pathway has an essential role in mammalian reproductive function and hormone production [43]. Variation in methylation patterns of genes such as *GNRHR*, *MCHR2*, and *PRP3* and the pathways they are involved in could be responsible for variation in growth, development, and responses to environmental stressors. The promoter region for *SYN4* was variable across all tissues and leukocytes. This gene is expressed in the brain, and the product of *SYN4* plays a vital role in dopamine release [44]. Expression levels of *SYN4* have been observed to have an inverse relationship with the methylation levels of the gene [45]. The variable gene expression of *SYN4* due to DNA methylation could result in differences in behavior. 

In leukocytes, numerous variable methylated features were identified to be involved in immune system response. Pathways such as the inflammation mediated by chemokine and cytokine signaling pathway are important for directing and controlling the migration of immune cells within the body, and shifts in gene expression can result in altered cytokine secretion. Similar results were found in peripheral blood monocytes in humans [7]. From studies in human monozygotic twins, there is an abundance of variable methylated loci around and within genes that are important for immune response [46]. In neutrophils, it is hypothesized that hypervariable sites are essential in establishing immune system response [47]. Variable methylation in the genes active in the immune system could lead to gene expression differences and influence immune response [48]. 

Many of the CpG sites with a β value SD ≥ 0.1 were associated with short and long interspersed retrotransposable elements and other highly repetitive regions of the genome (Figure 5). This is comparable to what Chatterjee and colleagues [6] found in methylation profiles of human neutrophils. Differences in DNA methylation patterns of long interspersed retrotransposable elements have been associated with low and high birthweights in humans [49]. The variable methylation patterns in sites within these elements could lead to phenotypic differences. All tissues had sites with β value SD ≥ 0.1 within the genomic regions that code for rRNA. Genomic sequences that code for rRNA exhibit substantial inter-individual variation, which can influence their methylation status [50].

The sites and features identified to have high variation in DNA methylation were mainly specific to tissue or leukocytes. Hannon and colleagues [51,52] observed similar patterns when comparing variation in DNA methylation patterns between whole blood and regions of the brain. In general, variation between tissues has been found to significantly exceed the inter-individual variation within a single tissue [53]. These results were consistent with those of Liu et al. [22], and suggested that hypervariable methylated regions likely harbor tissue-specific expressed genes. The variability in methylation patterns observed between these tissues and leukocytes potentially contributes to tissue-specific gene expression. The results suggest that using prediction of methylation patterns and variability across tissues may not be feasible. 

While the magnitude and distribution of CpG sites with high standard deviations were similar between the PNS and Control groups, only a slight overlap in the location of the CpG sites was observed in the anterior pituitary and leukocytes. Few of these were features identified to be highly variable in both groups. This could mean that prenatal stress in cattle influences the degree of variation in DNA methylation. There was a high proportion of sites with a *p* ≤ 0.001 for the χ^2^ test for the variance shared between the PNS and Control groups. The environmental factor rarely acts alone to influence the inter-individual variation in DNA methylation patterns [54]. Interaction between the environment and the cow genotype could be responsible for the differences in location of the inter-individual variation. 

Liu and colleagues [22] investigated the inter-individual variation of 28 semen samples from Holstein bulls. Highly variable methylation haplotypes were determined by comparing the standard deviation of the methylation levels of each region to the median standard deviation using the $\chi^{2}$ test for variance. There were 1681 highly variable methylated regions identified. While the highly variable regions constituted only 5.69% of total tested, many highly variable methylated regions between individuals were associated with key regulatory areas of gene expression. Numerous methylated regions were associated with reproduction traits and genomic regions [22]. These results provided novel insights into the contribution of natural DNA methylation variation to complex traits that are important to cattle productivity and health. 

This study provides the first characterization of inter-individual variation DNA methylation patterns in mature Brahman females across neural and endocrine tissues and leukocytes. These regions are important to consider for multiple reasons. Quantifying the inter-individual variation present is essential for future statistical analyses and interpretation of how a treatment or stressor affects DNA methylation at different genomic sites. These hypervariable regions in the genome could be linked to important genes or regulatory regions that contribute to complex performance traits and health in cattle.

The classical quantitative genetic decomposition of the phenotype consists of the influences of genotype, the environment and the interaction between the two. However, epigenetic mechanisms represent an additional source of differences among phenotypes. Variation in epigenetic marks such as DNA methylation can help explain a portion of phenotypic variation that cannot be explained by genetic differences. Understanding how DNA methylation contributes to differences in phenotype will be beneficial for future selection methods in beef cattle production systems. Characterization of the interaction of these variably methylated regions with gene expression and the environment may provide useful insight into cow performance in a variety of economically relevant metrics for calf production.

## 5. Conclusions

Analyses of the anterior pituitary gland, leukocytes, and amygdala revealed a small portion of various genomic features and CpG sites that contained highly variable DNA methylation patterns in mature cows that had experienced differing prenatal conditions. The PNS and Control groups had high variation between samples in DNA methylation patterns of the gene and promoter regions involved in behavior, hormone concentration, and immune response. Inter-individual variation within these genes could potentially contribute to differences in phenotype and performance in cattle, with potential consequences for overall animal health. Discordance in DNA methylation patterns between tissues is expected and common. The minimal overlap between pairs of tissues and leukocytes observed in this study also suggests that variability in DNA methylation patterns is tissue specific. The group of cows exposed to prenatal stress exhibited a similar number of variable methylated sites and features to the control group; the number of variable features and sites shared was small. Each tissue differed in the amount of variable methylated features, which could be due to the cell type heterogeneity of the sample. Use of emerging technologies for methylation profile analyses of single cells could enhance future studies by reducing the noise caused by the heterogeneity of cell types within a sample, and which contributes to observed variation. 

## Figures and Tables

**Figure 1 biology-12-00252-f001:**
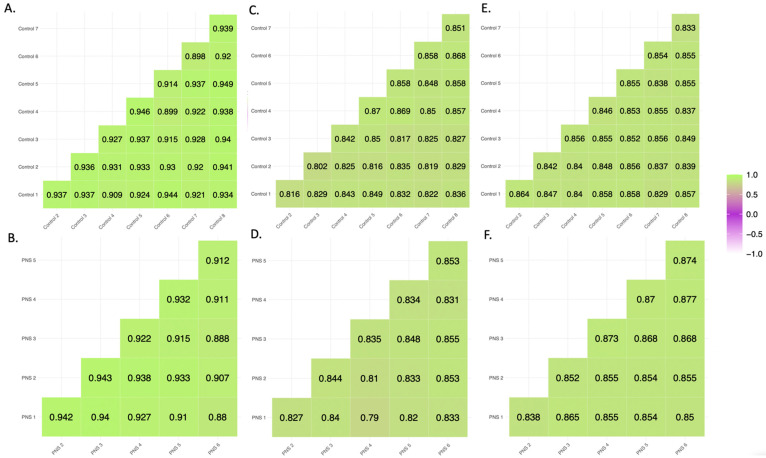
Pearson correlation coefficient estimates of samples (animals) using the beta values of cytosine-phosphate-guanine sites across the genome in the (**A**) Control amygdala, (**B**) Prenatally Stressed (PNS) amygdala, (**C**) Control anterior pituitary (**D**) PNS anterior pituitary (**E**) Control leukocytes, and (**F**) PNS leukocytes.

**Figure 2 biology-12-00252-f002:**
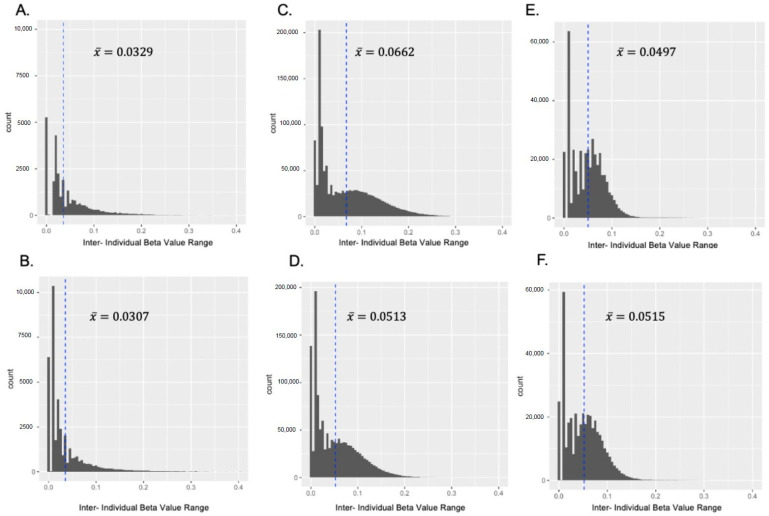
Histograms of the inter-individual beta value ranges for the cytosine-phosphate-guanine sites across the genome in the (**A**) Control amygdala, (**B**) Prenatally Stressed (PNS) amygdala, (**C**) Control anterior pituitary (**D**) PNS anterior pituitary (**E**) Control leukocytes, and (**F**) PNS leukocytes. The dashed lines represent the mean inter-individual beta value ranges.

**Figure 3 biology-12-00252-f003:**
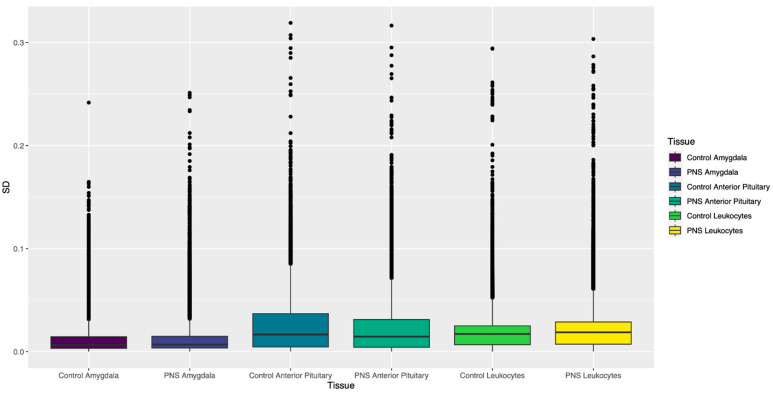
Box plots of standard deviations (SD) for the cytosine-phosphate-guanine sites (genome wide) analyzed in each tissue and leukocytes from the Prenatally Stressed (PNS) and Control groups.

**Figure 4 biology-12-00252-f004:**
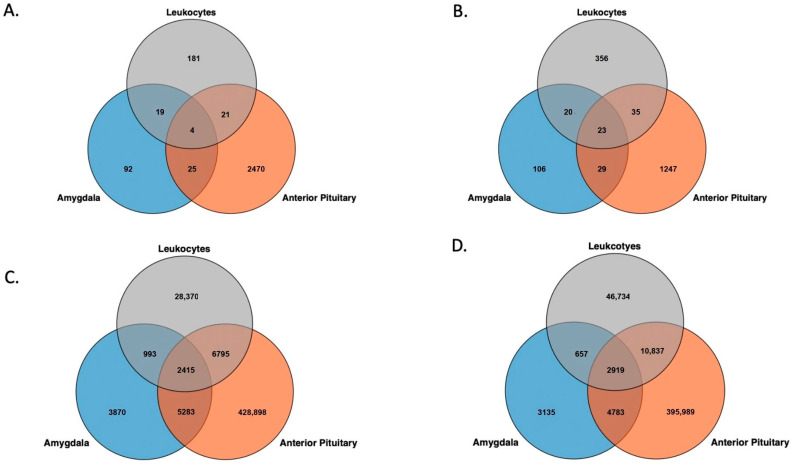
Overlap of genome wide cytosine-phosphate-guanine sites with a beta value standard deviation ≥0.1 across the tissues and leukocytes in the (**A**) Control and (**B**) Prenatally Stressed group and cytosine-phosphate-guanine sites with a *p* ≤ 0.001 for the chi-square test for the variance in the (**C**) Control and (**D**) Prenatally Stressed group.

**Figure 5 biology-12-00252-f005:**
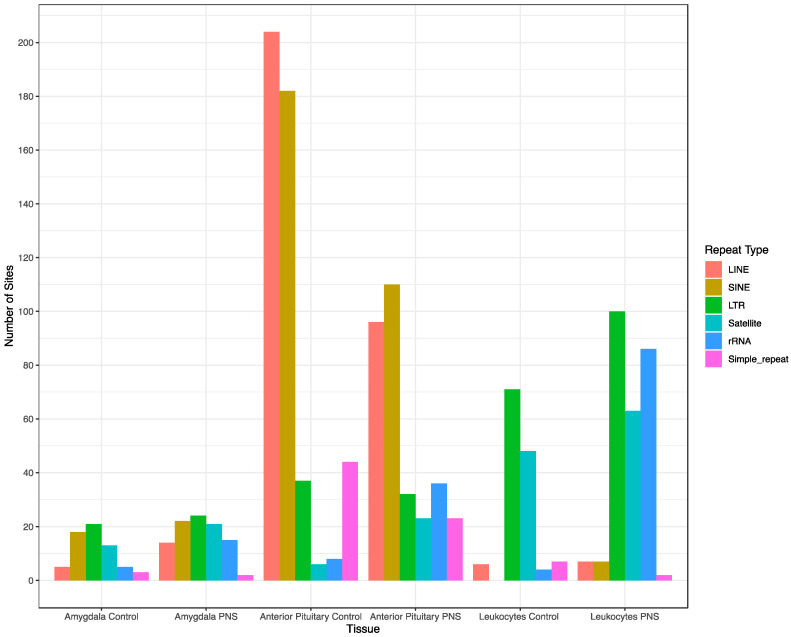
Number of sites with a beta value standard deviation ≥ 0.1 associated with different types of repetitive elements, long interspersed nuclear elements (LINE), short interspersed nuclear elements (SINE), long terminal repeat (LTR), satellite regions, ribosomal RNA (rRNA) and simple repeats throughout the genome.

**Figure 6 biology-12-00252-f006:**
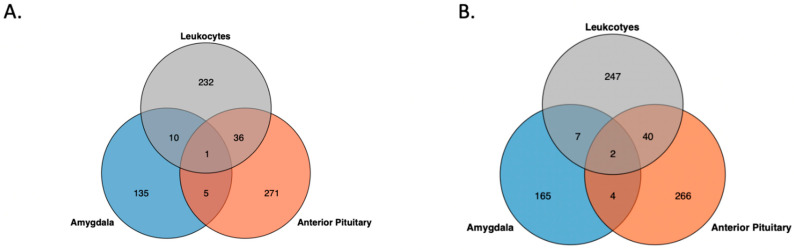
Overlap of gene bodies with high DNA methylation variation across the tissues and leukocytes in the Control (**A**) and Prenatally Stressed (**B**) group.

**Table 1 biology-12-00252-t001:** Number of cytosine-phosphate-guanine with high variability in DNA methylation within treatment groups and tissues.

	SD ≥ 0.1 ^1^	Mean Variance ^2^	*p* ≤ 0.001 ^3^
Amygdala			
Control	140	2.76 × 10^−4^	12,471
Prenatally Stressed	178	4.81 × 10^−5^	11,494
Anterior Pituitary			
Control	2520	4.11 × 10^−5^	443,391
Prenatally Stressed	1334	2.19 × 10^−4^	414,528
Leukocytes			
Control	225	2.95 × 10^−4^	38,573
Prenatally Stressed	434	3.26 × 10^−4^	61,147

^1^ Number of sites with a beta value standard deviation greater than 0.1. ^2^ Mean variance of the beta values at site. ^3^ Number of sites that had beta value variance statistically greater (χ^2^ test for the variance) than the mean variance at an *FDR* ≤ 0.001.

**Table 2 biology-12-00252-t002:** The number of genomic features with high variability in DNA methylation between cows in the Control group.

	Total Features Tested ^1^	Significant ^2^	% of Total Features Tested ^3^
Amygdala			
Promoter ^4^	13,053	52	0.40%
Gene	16,828	151	0.90%
CpG ^5^ islands	16,808	51	0.50%
CpG shores ^6^	18,754	81	0.43%
Leukocytes			
Promoter	15,602	168	0.94%
Gene	18,829	281	1.49%
CpG islands	20,016	216	0.01%
CpG shores	17,417	149	0.61%
Anterior Pituitary			
Promoter	20,688	226	1.09%
Gene	19,083	314	1.65%
CpG islands	20,605	307	1.49%
CpG shores	30,252	446	1.47%

^1^ Number of genomic features tested after removal of features with null methylation values. ^2^ Number of genomic features with adjusted *p* ≤ 0.05 for the variance intensity difference test. ^3^ Percent of the total number of features with adjusted *p* ≤ 0.05. ^4^ Promoter regions were defined as 1000 bp upstream of the transcription start site of a gene, and 500 bp downstream of the transcription start site. ^5^ Cytonsine-phosphate-guanine. ^6^ CpG shores were defined as 2000 bp upstream and 2000 bp downstream of CpG islands.

**Table 3 biology-12-00252-t003:** The number of genomic features with high variability in DNA methylation between cows in the Prenatally Stressed group.

	Total Tested ^1^	Significant ^2^	% of Total Features Tested ^3^
Amygdala			
Promoter ^4^	13,177	52	0.40%
Gene	17,021	178	1.05%
CpG islands ^5^	16,700	51	0.31%
CpG shores ^6^	19,244	7	0.04%
Leukocytes			
Promoter	16,255	168	1.03%
Gene	19,290	298	1.54%
CpG islands	20,400	216	1.06%
CpG shores	29,987	149	0.50%
Pituitary Gland			
Promoter	20148	372	1.85%
Gene	18,965	313	1.65%
CpG islands	20,594	313	1.52%
CpG shores	29,675	399	1.34%

^1^ Number of genomic features tested after removal of features with null methylation values. ^2^ Number of genomic features with an adjusted *p* ≤ 0.05 for the variance intensity difference test. ^3^ Percent of the total number of features with an adjusted *p* ≤ 0.05. ^4^ Promoter regions were defined as 1000 bp upstream of the transcription start site of a gene and 500 bp downstream of the transcription start site. ^5^ Cytosine-phosphate-guanine. ^6^ CpG shores were defined as 2000 bp upstream and 2000 bp downstream of CpG islands.

**Table 4 biology-12-00252-t004:** The numbers of genomic features that had significant variation in DNA methylation in both the Prenatally Stressed and Control groups.

Feature	Anterior Pituitary	Amygdala	Leukocytes
Promoter ^1^	92	1	14
Gene	81	6	54
CpG ^2^ islands	73	0	26
CpG shores ^3^	96	0	3

^1^ Promoter regions were defined as 1000 bp upstream of the transcription start site of a gene and 500 bp downstream of the transcription start site. ^2^ Cytosine-phosphate-guanine. ^3^ CpG shores were defined as 2000 bp upstream and 2000 bp downstream of CpG islands.

**Table 5 biology-12-00252-t005:** Highly variable ^1^ genomic features involved in the immune response biological process ^2^.

Feature Name	Feature Type	Standard Deviation ^3^
** Prenatally Stressed group **		
ENSBTAG00000020813	Promoter ^4^	32.97
WAP four-disulfide core domain 2	Promoter	27.23
ENSBTAG00000014329	Gene	27.19
ENSBTAG00000052841	Gene	26.61
ENSBTAG00000036102	Gene	26.02
Interleukin-34	Promoter	24.99
Secretory leukocyte peptidase inhibitor	Promoter	24.6
Bovine major histocompatibility complex	Promoter	24.29
Interleukin 9 receptor	Promoter	23.86
Myelin oligodendrocyte glycoprotein	Promoter	22.72
Transmembrane protein 176B	Gene	21.44
ENSBTAG00000048980	Gene	20.35
Interleukin 36 alpha	Gene	18.73
ENSBTAG00000023563	Gene	18.26
C-X-C chemokine receptor type 2	Gene	18.13
Triggering receptor expressed on myeloid cells 1	Gene	17.94
Testicular cell adhesion molecule 1	Gene	17.32
Interleukin-4	Gene	16.9
ENSBTAG00000006864	Gene	16.48
** Control group **		
Myelin oligodendrocyte glycoprotein	Promoter	37.67
ENSBTAG00000045810	Gene	33.86
ENSBTAG00000020813	Promoter	33.04
ENSBTAG00000055111	Promoter	30.69
ENSBTAG00000006864	Gene	29.81
Peptidase inhibitor 3	Gene	28.62
Interleukin-34	Promoter	26.04
ENSBTAG00000051008	Promoter	25.91
Calcium-dependent phospholipase A2	Gene	24.7
ENSBTAG00000050878	Gene	24.28
Interleukin 2 receptor subunit beta	Promoter	23.78
CCAAT enhancer binding protein epsilon	Gene	22.53
ENSBTAG00000053521	Gene	22.38
Major histocompatibility complex, class I-related	Gene	19.45
5,-aminolevulinate synthase 2	Gene	19.27
C-C motif chemokine ligand 25	Gene	18.83
C-C motif chemokine ligand 1	Gene	17.97

^1^ Variance intensity difference statistical test *FDR* ≤ 0.05. ^2^ Biological process identified through protein analysis through evolutionary relationships analysis. ^3^ Standard deviation of mean methylation of defined genomic features. ^4^ Promoter regions were defined as 1000 bp upstream of the transcription start site of a gene and 500 bp downstream of the transcription start site.

## Data Availability

The data that support the findings of this study are available from the corresponding author, DGR, upon reasonable request.

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
