# Peer review of "Inter-Individual Variation in DNA Methylation Patterns across Two Tissues and Leukocytes in Mature Brahman Cattle"

_biology, 2023, doi:10.3390/biology12020252_

Round 1

Reviewer 1 Report

Please refer to the below comments:

 1.      Lines 107 – 113: There were 39 cows in this project (21 for PNS and 18 for Control); why the authors used only 14 cows (6 for PNS and 8 for Control) for investigation? In addition, the numbers of selected animals for each group were unequal. The explanations are needed to provide.

2.      Lines 100 – 113: Only female animals were used, so what was the hypothesis of different methylation patterns in the cow calve production system?

3.      Line 214: Please provide IBR values for 2C and 2D.

4.      Figure 2: It would be better for the readers if the authors added mean IBR values for each blue line.

5.      Figure 3: Please add a space between “Control Anterior Pituitary” and “PNS Anterior Pituitary”

6.      Line 449 – 452: Please provide suggestions on how to use these findings to manage or what should be concern in the cow calve production system.

Author Response

  1. Lines 107 – 113: There were 39 cows in this project (21 for PNS and 18 for Control); why the authors used only 14 cows (6 for PNS and 8 for Control) for investigation? In addition, the numbers of selected animals for each group were unequal. The explanations are needed to provide.
    1. The 39 heifer calves entered the production system where they were exposed to a bull annually and culled for reproductive failures the subsequent years. At the time of investigation, 14 of the remaining cows (not pregnant) were harvested for analysis.
    2. The farm where cows were maintained is 3 hours from the campus slaughter facility. In error, farm personnel sent a pregnant prenatally stressed female and an extra Control female instead of the designated PNS female. We decided to send both cows back to the farm which resulted in only 6 prenatally stressed cows slaughtered.
  2. Lines 100 – 113: Only female animals were used, so what was the hypothesis of different methylation patterns in the cow calve production system?
    1. Addition to line 89: “The methylome has the potential to influence important production aspects such as susceptibility to disease, adaptability to stressors such as heat stress, fertility and even restricted feed intake. Variability in DNA methylation patterns between cows could result in performance differences that impact profit.”
  3. Line 214: Please provide IBR values for 2C and 2D.
    1. Line 214 inserted: “Control 0.0662 and PNS 0.0513”
  4. Figure 2: It would be better for the readers if the authors added mean IBR values for each blue line.
    1. Means corresponding to each blue line were added in Figure 2.
  5. Figure 3: Please add a space between “Control Anterior Pituitary” and “PNS Anterior Pituitary”
    1. Figure 3 was adjusted so that there is a space “Control Anterior Pituitary” and “PNS Anterior Pituitary”.
  6. Line 449 – 452: Please provide suggestions on how to use these findings to manage or what should be concern in the cow calve production system.
    1. Addition at line 449: “The classical quantitative genetic decomposition of phenotype consists of the influences of genotype, the environment and the interaction between the two. However, epigenetic mechanisms represent an additional source of differences among phenotypes. Variation in epigenetic marks such as DNA methylation can help explain a portion of phenotypic variation that could not be explained by genetic differences. Understanding how DNA methylation contributes to differences in phenotype will be beneficial for future selection methods in beef cattle production systems. Characterization of the interaction of these variably methylated regions with gene expression and the environment may give useful insight into cow performance in a variety of economically relevant metrics for cow calf production”

Reviewer 2 Report

This is an interesting and well-written manuscript that utilized methylation data from mature Brahman females to quantify the inter-individual variation in DNA methylation. The methodology is described with sufficient detail. Results are clear but some figures were not described. Discussion is well supported. Conclusions are interesting. References writing and citations should be corrected.

I suggest considering next minor comments to improve the manuscript:

-       According to guidelines described in the “Instructions for Authors” section, all references within the text must be numbered in orden of appearance, and these reference numbers should be placed within square brackets.

-       In References section, all references should be corrected following the guidelines described in the “Instructions for Authors”.

-       In Results section, the figures 1A, 1B, 2E and 2F were not described within the text.

-       Conclusions section is too long, I suggest to the authors that consider if the second paragraph could be incorporated in Discussion section.

Author Response

This is an interesting and well-written manuscript that utilized methylation data from mature Brahman females to quantify the inter-individual variation in DNA methylation. The methodology is described with sufficient detail. Results are clear but some figures were not described. Discussion is well supported. Conclusions are interesting. References writing and citations should be corrected.

I suggest considering next minor comments to improve the manuscript:

  1. According to guidelines described in the “Instructions for Authors” section, all references within the text must be numbered in order of appearance, and these reference numbers should be placed within square brackets.
    1. In text citations and references have been adjusted to follow the “Instruction for Authors”
  2. In Results section, the figures 1A, 1B, 2E and 2F were not described within the text.
    1. Addition to line 203: “Figures 1A and B”
    2. Addition to line 209: “closely followed by the mean IBRs from the leukocytes (Figures 2E &F).”
  3. Conclusions section is too long, I suggest to the authors that consider if the second paragraph could be incorporated in Discussion section.
    1. Lines 471-479 were removed from the conclusion section.
    2. Information was incorporated into the discussion section lines 450-458.

Reviewer 3 Report

Comments to the Authors 
In this manuscript, the authors aim to investigate the changes of DNA methylation and focus on gene variation of the for prenatal stress in two tissue, amygdala and pituitary, and leucocyte. However, I have serious concerns over the use of these statistics data without phenotype of Brahman cow, and I believe some of the conceptual models underlying the authors' approach to be flawed. Moreover, the data of methylation analyzes have been lost relevance to indicate prenatal stress, because there are no empirical studies confirming to the effects on cow.

Specific points to be addressed are as follows: 

Major points

1.     The experimental aim and methods section did not clearly explain why the authors set up this experiment using two tissues, amygdala and pituitary, and leukocytes. I couldn’t find that insist on the experimental design of heat stress and transport stress for Brahman cow as advantage of animal models. 

2.     I recommend to see again some experimental design discussion of the findings of previous paper which is your reference in relation to recent findings in sperm DNA methylation in cattle. It was clearly understood because there was revealing the methylation region of several genes including QTL related gene and phenotype of sperm motility and reproduction. In this experiment, the author did not show the empirical research for prenatal stress of Brahman cows when it performed the analysis of methylation metadata.

3.     Pass way analysis and Biological function would be more important when the authors will provide the physiological phenotype of DNA methylation patterns and gene expression. I would like to advise to use the supplemental data which is writing Pass way analysis, interaction data, and candidate genes for perinatal stress. It is interesting to analyze of data for reveal the variation between samples in DNA methylation patterns for behavior, Hormone, and immune response.

Minor points

1.     Please indicate the overview of the study design and data analyses workflow of the sequencing data. 

2.     Figure 2 has been disappeared E and F.

3.     L.134 one mM EDTA>>1mM EDTA.

4.     L.140, L.142 TM are large. L.492,L.496, A and T are Bold.

5.     The graphs on Fig6 A, does it wrong figure?

Author Response

Major points

  1. The experimental aim and methods section did not clearly explain why the authors set up this experiment using two tissues, amygdala and pituitary, and leukocytes. I couldn’t find that insist on the experimental design of heat stress and transport stress for Brahman cow as advantage of animal models. 
    1. Prenatal transportation stress has been demonstrated to alter phenotypes and induce physiological responses including significantly increased vaginal temperature, shrink, and serum concentrations of cortisol and glucose in cows of earlier projects and in the present project (Lay et al.,1997, Littlejohn et al., 2018 Price et al., 2015; all cited in the manuscript). Stressed induced epigenetic changes could be responsible for the performance differences in prenatally stressed offspring.
    2. The amygdala and anterior pituitary gland both have active roles within the brain for hormone and stress response. The leukocytes were used as a biological sample not within the brain.
      1. Addition to line 113: “The amygdala and anterior pituitary gland were chosen because of their importance in hormone production and stress response. Leukocytes were chosen for the analysis as a sample set originating from outside of the brain”
    3. Addition to line 102: “These dams exhibited increase vaginal temperature, shrink, and increased serum cortisol and glucose in response to the transportation events confirming a physiological stress response (Price et al., 2015).”
  2. I recommend to see again some experimental design discussion of the findings of previous paper which is your reference in relation to recent findings in sperm DNA methylation in cattle. It was clearly understood because there was revealing the methylation region of several genes including QTL related gene and phenotype of sperm motility and reproduction. In this experiment, the author did not show the empirical research for prenatal stress of Brahman cows when it performed the analysis of methylation metadata.
    1. Such empirical work investigating DNA methylation patterns in Brahman cattle was reported in Littlejohn et al. (2018), Baker et al. (2020), and Cilkiz et al. (2021). The objectives of the present work were developed using those results as a foundation. Methylation of cytosines were not identical across cows within (or across) treatments.  The between individual variability has been reported in humans (Chatterjee et al., 2015) and in bovine sperm (Liu et al., 2018); however, it would not be expected that the variation observed in either of those would adequately portray the inter-individual variation in bovine somatic cell lines.
    2. Addition to line 363: “Methylation patterns in prenatally stressed Brahman cattle was first presented by Littlejohn et al. [17]), Baker et al. [18], and Cilkiz et al. [19]. In leukocytes harvest at 28 days of age of age from the same animals described in this project there was vast differences in DNA methylation patterns between the prenatally stressed group and the Control. Many of the differentially methylated cytosines were located within regulatory genes active in hormone production, immune response, and development [17-18]. Differences in methylation of the leukocytes between the two groups diminished between 28 days and 5 years of age [19]. Methylation patterns varied within each group at both time periods. The between individual variability has been reported in humans [6] and in bovine sperm [22]. However, methylation patterns differ greatly between species, sexes, and tissue, suggesting the variation observed in either would not adequately portray the inter-individual variation in bovine soma cells.”
  3. Pass way analysis and Biological function would be more important when the authors will provide the physiological phenotype of DNA methylation patterns and gene expression. I would like to advise to use the supplemental data which is writing Pass way analysis, interaction data, and candidate genes for perinatal stress. It is interesting to analyze of data for reveal the variation between samples in DNA methylation patterns for behavior, Hormone, and immune response.
    1. Baker et al. (2022) reported gene expression and methylation results. The objective of this study was to analyze the inter-individual variation in DNA methylation patterns in Brahman cattle independent of gene expression or phenotypes observed.  We considered it reasonable that regions with an excess of variability in one treatment group (but not in the other) might constitute another opportunity to identify important gene and pathway differences between treatments and identify potential biological mechanisms that could be affected.

Minor Points:

  1. Please indicate the overview of the study design and data analyses workflow of the sequencing data. 
    1. Inserted “HiSeq” on line 146 to clarify the type of Illumina technology used for sequencing
  2. Figure 2 has been disappeared E and F.
    1. Figure 2 was adjusted so that E and F are now present in the figure.
  3. 134 one mM EDTA>>1mM EDTA.
    1. “1” changed to one in line 135
  4. 140, L.142 TM are large. L.492,L.496, A and T are Bold.
    1. TM has been changed to the proper superscript denotation.
  5. The graphs on Fig6 A, does it wrong figure?
    1. Figure 6 has been adjusted so that figure 6A has color.

Round 2

Reviewer 3 Report

The authors have been modified the manuscript to reviewers comments. Please ensure to write an animal ethics for this experiment using animals.

Author Response

An animal ethics and care statement was provided. Addition to line 109: All procedures were done in compliance with the Guide for the Care and Use of Agricultural Animals in Research and Teaching [23], and its earlier versions, and approved by the Texas A&M AgriLife Research Animal Care and Use Committee